# Study protocol for investigating the performance of an automated blood test measuring GFAP and UCH-L1 in a prospective observational cohort of patients with mild traumatic brain injury: European BRAINI study

Marion Richard,[1] Alfonso Lagares [ID],[2] Victor Bondanese,[3] Javier de la Cruz,[4] Odile Mejan,[3] Vladislav Pavlov,[5] Jean-François Payen,[1] BRAINI investigators

For numbered affiliations see end of article.

**Correspondence to**
Dr Alfonso Lagares; alfonlag@ucm.es

## ABSTRACT

**Introduction** Mild traumatic brain injury (mTBI) is a common cause of clinical consultation in the emergency department. Patients with mTBI may undergo brain CT scans based on clinical criteria. However, the proportion of patients with brain lesions on CT is very low. Two serum biomarkers, glial fibrillar acidic protein (GFAP) and ubiquitin carboxy-terminal hydrolase L1 (UCH-L1), have been shown to discriminate patients regarding the presence or absence of brain lesions on initial CT scan when assessed within the first 12 hours after TBI. However, the current technique for measuring serum concentrations of GFAP and UCH-L1 is manual and time consuming, which may hinder its use in routine clinical practice. This study assesses the diagnostic accuracy of an automated assay for the measurement of serum GFAP and UCH-L1 in a cohort of patients with mTBI who received a CT scan as the standard of care.

**Methods and analysis** This is a prospective multicentre observational study of 1760 patients with mTBI recruited in France and Spain across 16 participating sites. Adult patients with an initial Glasgow Coma Scale score of 13–15 and a brain CT scan underwent blood sampling within 12 hours after TBI. The primary outcome measure is the diagnostic performance of an automated assay measuring serum concentrations of GFAP and UCH-L1 for discriminating between patients with positive and negative findings on brain CT-scans. Secondary outcome measures include the performance of these two biomarkers in predicting the neurological status and quality of life at 1 week and 3 months after the trauma.

**Ethics and dissemination** Ethics approval was obtained by the Institutional Review Board of Sud-Ouest Outre Mer III in France (Re#2019-A01525-52) and Hospital 12 de Octubre in Spain (Re#19/322). The results will be presented at scientific meetings and published in peer-reviewed publications.

**Trial registration number** ClinicalTrials.gov: NCT04032509.

---

### Strengths and limitations of this study

► This is a large prospective multicentre study to validate the value of serum biomarkers glial fibrillar acidic protein and ubiquitin carboxy-terminal hydrolase-L1 in predicting brain CT-scan findings after mild traumatic brain injury (mTBI).

► The additional value of these two biomarkers will be assessed regarding neurological outcome and quality of life.

► Variability in the management of patients with mTBI and in the indications of CT scans may influence the rate of positive CT scans in the different participating centres, which may have an impact on the statistical power of the study.

---

## INTRODUCTION

Mild traumatic brain injury (mTBI), as defined by a Glasgow Coma Scale (GCS) score of 13–15,[1] represents a frequent admission in the emergency department (ED).[2–4] The initial management includes a non-contrast brain CT scan if the patient meets specific conditions. However, the prevalence of CT-detected abnormalities is less than 10% among patients with mTBI, and less than 1% of them will require neurosurgical procedures. Efforts have been made for decades to optimise the indications for brain CT scans after mTBI such as the New Orleans Criteria[5] and Canadian Head CT Rule[6] as well as national guidelines such as the French guidelines.[7] However, a certain variability exists among physicians regarding CT scan indications and some situations post-TBI may be confusing, for example, intoxicated patients or patients with hearing loss or speech disturbances. As a consequence, up to 40% of CT

scans prescribed in EDs would actually do not follow guideline recommendations,[8] reflecting a substantial CT overuse.

Clinical decision rules for initial CT scan can be optimised with the use of an objective parameter that is easily and rapidly assessed. This could be achieved using blood concentrations of brain-damage biomarkers. Among candidates, serum protein S100B is the only biomarker used in Europe. The Scandinavian guidelines for the initial management of mild and moderate head injuries in adults provide recommendations for the use of S100B to rule out the need for head CT in mTBI. Multicentre validation of Scandinavian Guidelines is currently in progress in Sweden (*ClinicalTrials.gov, 2017, NCT03280485*). Unfortunately, the blood concentration of S100B can be affected by several factors such as multiple trauma, skin colour, presence of melanomas, and its clinical utility is limited by the short half-life (3 hours) of S100B in blood.[9]

Recently, serum levels of ubiquitin carboxy-terminal hydrolase-L1 (UCH-L1) and glial fibrillar acidic protein (GFAP), two brain-specific proteins, were found to be elevated in patients with intracranial lesions visible on CT scans.[10 11] Data from the Evaluation of Biomarkers of Traumatic Brain Injury (ALERT-TBI) trial showed that serum GFAP and UCH-L1 protein concentrations are able to reliably predict the absence of clinically relevant lesions on CT scan in patients with mTBI.[12] The Banyan Brain Trauma Indicator (*Banyan BTI Package Insert*) has obtained FDA-clearance in February 2018. This is a manual immunoassay that measures GFAP and UCH-L1 serum concentrations with a sensitivity (95% lower confidence limit) and negative predictive value (NPV; 95% lower confidence limit) of 97.5% (93.7%) and 99.6% (98.8%), respectively.[13] However, the manual ELISA technique takes 4 hours to provide results, which is too long for mTBI triage in the ED setting. In addition, the cut-offs used in the ALERT-TBI trial need to be externally validated.[12 14]

A faster in vitro diagnostic (IVD) technique is then required for a possible use in clinical practice, together with an external validation of the diagnostic accuracy of GFAP and UCH-L1 in patients with mTBI.[15–18] Automated assays assessing serum concentrations of GFAP and UCH-L1 have been developed on the VIDAS platform (bioMérieux, Marcy l'Etoile, France). The primary objective of the study is to evaluate the diagnostic performance of the VIDAS GFAP and VIDAS UCHL-1 assays in a prospective multicentre cohort of patients with mTBI with respect to their brain CT scan findings. The secondary objectives are to assess the ability of the two biomarkers to predict the neurological status and quality of life at 1 week and 3 months after mTBI.

## METHODS AND ANALYSIS
### Study design
BRAINI is a prospective, multicentre, observational study in France and Spain.

### Study setting
BRAINI includes 12 sites in France within university hospitals (Grenoble, Lyon Edouard-Herriot, Lyon-Sud, Tours, Nantes, Dijon, Poitiers, Montpellier, Toulouse and Bordeaux) and non-university hospitals (Annecy and Villefranche-sur-Saone) and 4 sites in Madrid, Spain, including University Hospital 12 de Octubre, University Hospital Gregorio Marañón, Hospital del Tajo and Hospital de La Princesa. Each centre was chosen based on documentation with regard to patient availability and experience in mTBI patient management.

### Study population
Patients will be included if they meet the following criteria: age >18 years (France) and >15 years old (Spain), admitted for a mTBI with GCS score 13–15, requiring brain CT scan as part of standard of care according to the French guidelines[7] or to the in-charge physician in Spain, and 10 mL blood sample obtained as part of routine blood samples within 12 hours after injury.

Patients will be excluded if they have at least one of the following criteria: GCS score 3–12 on admission; time of injury unknown; time since injury exceeding 12 hours; primary admission for non-traumatic neurological disorder (eg, stroke, spontaneous intracranial haematoma); penetrating head injury; mechanical ventilation; neuropsychiatric and neurological comorbidities that might interfere with the assessment of outcomes at 1 week and 3 months; venepuncture not feasible; no brain CT scan; subject under judiciary control; pregnant or breast-feeding woman; or participation in another therapeutic study.

Patients will be included after verification of the eligibility criteria. In France patient non-opposition to participate in the study must be documented. In Spain, written informed consent will be obtained, before inclusion in the study, from the patient or next of kin if the patient is not in condition of giving consent.

### Study outcomes
The primary outcome measure is the performance of the VIDAS GFAP and VIDAS UCHL-1 assays in terms of sensitivity, specificity, positive predictive value, NPV and their corresponding lower limit of the 95% CI with respect to brain CT scan findings, that is, positive versus negative (see below).

Secondary outcomes are measured at 1 week and 3 months post TBI and include neurological status, that is, stable or degraded condition, and quality of life assessed using the Quality of Life after Brain Injury (QOLIBRI) questionnaire (quality of life after TBI)[19–21] at 1 week. At 3 months post TBI, patients will be assessed according to the Extended Glasgow Outcome Scale,[22 23] the 5-level European Quality of Life-5 Dimensions (EQ-5D) version,[24] the QOLIBRI scale, and the Rivermead Post-concussion Symptoms Questionnaire.[25] The study design and flow is shown in figure 1.

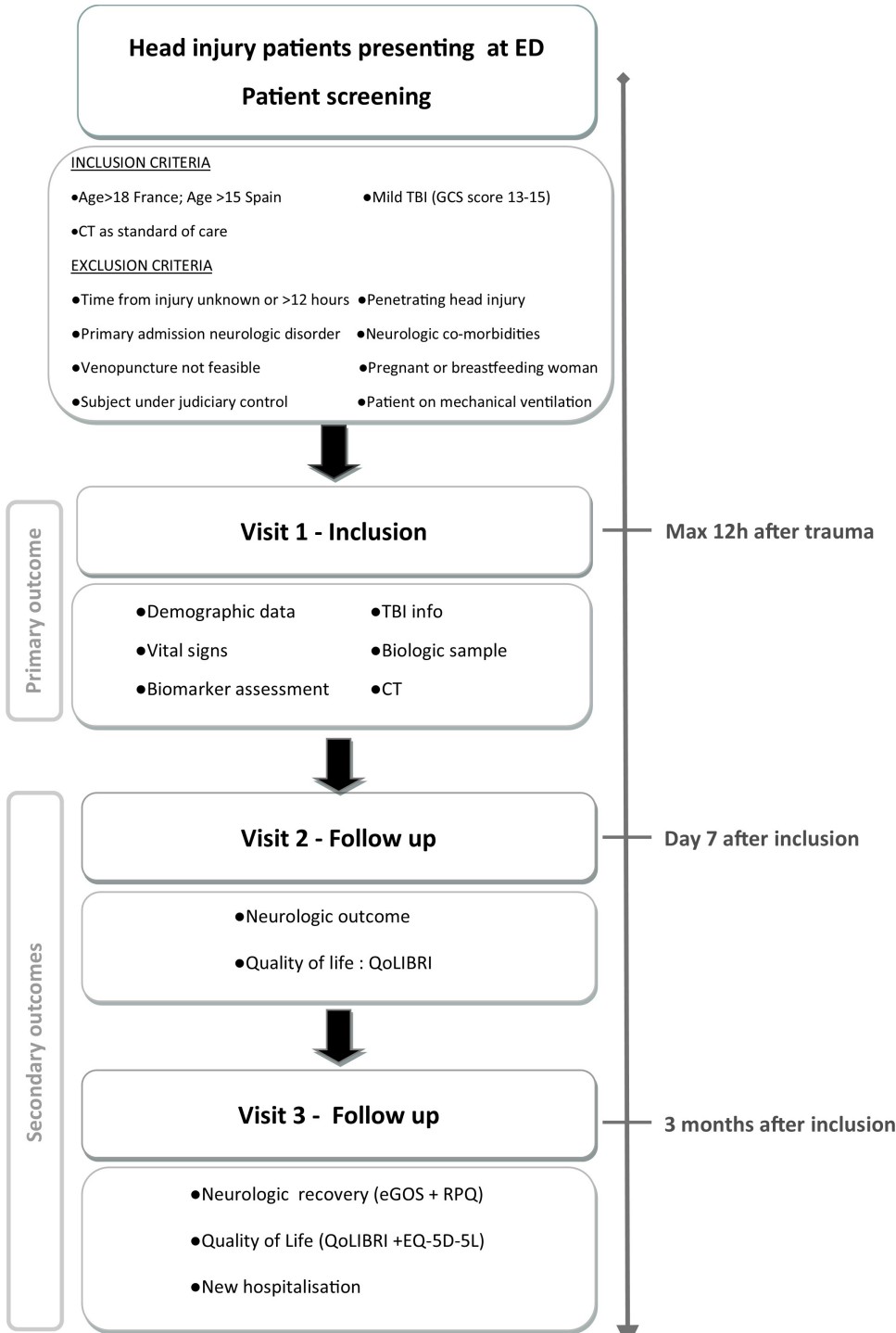

**Figure 1** Study design and flow of the BRAINI study. ED, emergency department; eGOS, extended Glasgow Outcome Scale; EQ-5D, European Quality of Life-5 Dimensions; GCS, Glasgow Coma Scale; QOLIBRI, Quality of Life after Brain Injury; RPQ, Rivermead Post-concussion Symptoms Questionnaire; TBI, traumatic brain injury.

## Data collection and data monitoring

At each participating site, data will be collected and entered into the web-based electronic CRF (eCRF) (Medsharing, Fontenay-sous-Bois, France) by clinical research associates under the supervision of the site principal investigators. The study database will be created from the eCRF. Trained research coordinators will monitor the data collection. The study will collect demographic and baseline information at admission, the reason to prescribe brain CT scan and immediate CT findings by the local radiologist, biological data if indicated by the in-charge physician, a 10 mL study-specific blood sample within 12 hours after trauma, and functional outcomes at 1 week and 3 months. At 1 week post TBI, clinical research associates will transmit information regarding the neurological status of the patients to

the coordinating centres (University Hospital Grenoble Alpes and University Hospital 12 de Octubre, Madrid). Questionnaires at 3 months will be conducted by trained central outcome assessors during a structured telephone interview. Anonymized data from brain CT images will be transferred to a centralised platform to be evaluated by a trained neuroradiologist. Capture of all data, particularly the functional outcome assessments and CT scan analysis, will be conducted by persons who are blinded to the results of the two biomarkers.

### Data analysis: CT scan

Brain CT scans are performed as part of the patient standard of care and uploaded to a secure web central database (SHANOIR-INRIA) for archive and central analysis. CT scan findings are classified as CT negative or CT positive by the local and one central reading. In case of disagreement between them, an additional central CT reading will be performed by a third radiologist for final adjudication. The criteria for CT positivity are crucial because the primary objective is related to the CT results (positive/negative). Criteria for CT-positive findings are as follows: a CT scan classified as type II or greater according to the Traumatic Coma Databank classification (range I–V), or a CT scan with the presence of the following lesions: (1) epidural haematoma, (2) subdural haematoma, (3) subarachnoid haemorrhage, (4) intraventricular haemorrhage, (5) contusion, (6) petechial haemorrhage or (7) any finding related to diffuse axonal injury and depressed skull fracture. Linear skull fractures will be recorded but not used for the definition of CT positive lesions.

### Data analysis: biomarkers

The 10 mL blood sample for determination of biomarkers will be collected using gel-separator tubes and allowed to clot for 30 min at room temperature (18°C–25°C) and then centrifuged at 2000g for 15 min. Serum will be aliquoted to 1 mL cryovials, frozen and stored at −80°C within 2 hours of the extraction until shipment on dry ice to the central storage facility (bioMérieux R&D bioBank, Marcy l'Étoile, France). All measurement procedures will be conducted independently and blinded to the clinical and CT status of the patients. The samples will be batch analysed using fully automated instruments (VIDAS, bioMérieux) with kits to detect the serum concentrations of UCH-L1 and GFAP. bioMérieux will perform analytical performance according to Clinical & Laboratory Standards Institute guidelines including precision study, that is, repeatability and reproducibility of the assays. In the absence of reference standard and reference measurement procedure for GFAP and UCH-L1, the VIDAS assays will be traceable to an internal standard. The metrological traceability chain defined in accordance with the standard ISO 17511 will ensure the GFAP and UCH-L1 values assigned to calibrators, product calibrator and patient samples.

### Statistical considerations

The Banyan BTI test for which Banyan biomarkers got FDA clearance is the only IVD reference assay for GFAP and UCHL-1 so far. It will be used as the reference for the VIDAS GFAP and VIDAS UCHL-1 assays.

We formulate the hypothesis that the automated assay measuring the serum concentrations of GFAP and UCH-L1 will have a sensitivity of at least 97% and an NPV of at least 99% to rule out the need for CT scan with accuracy. Assuming a prevalence of 11% of positive brain CT scans, the enrolment of 176 patients with positive CT-scans and 1424 patients with negative CT-scans is required to achieve these indices of performance. Assuming a 10% rate of loss to follow-up, the study recruitment target is set to 1760 patients in France and in Spain.

Cut-off values for UCH-L1 (327 pg/mL) and GFAP (22 pg/mL) applied in the ALERT-TBI study were fixed for the Banyan BTI.[12] We will verify the predictive performances of UCH-L1 and GFAP measured on the VIDAS instrument with these predefined cut-off values. An analysis on whether alternative cut-offs could provide better performance with the VIDAS BTI will also be performed based on BRAINI study data. A method comparison between Banyan BTI and VIDAS BTI will be performed and will contribute to optimal cut-offs selection.

The distribution of biomarker levels will be analysed by age, gender and other predictors. As a secondary objective, the levels of GFAP and UCH-L1 obtained with the VIDAS assays will also be analysed according to the neurological outcome, the presence of posttraumatic symptoms and quality of life (secondary objective). To test whether these biomarkers are independent prognostic factors, multivariable analysis will be conducted using logistic regression for dichotomised variables and linear regression for quantitative variables. In addition, prediction models will be developed to assess the contribution of biomarkers to existing diagnostic and prognostic tools.

### Patient and public involvement

No patient involved.

### Ethics and dissemination

Ethics approval was obtained from the Comité Ético de Investigación Clínica of Hospital 12 de Octubre on 20 July 2019, Madrid (Re#19/322) and the study started in August 2019. In France, the protocol was approved by the Institutional Review Board of Sud-Ouest Outre Mer III on 14 November 2019 (Re#2019-A01525-52). The study began in France in Grenoble on 29 November 2019. The National Commission on Informatics and Liberty (France) gave its approval on 31 January 2020 (Re#919443).

The results of this study will be presented at national and international meetings and published in peer-reviewed journals. Patients will not be individually notified regarding the results of the study. The principal publication from the study will be in the name of the BRAINI investigators with full credit assigned to all active,

collaborating investigators, research coordinators and institutions.

## DISCUSSION

The protocol and future results of the current study should be analysed taking into account previously published data. In the large CENTER-TBI cohort study, performance of a panel of six blood biomarkers including UCH-L1 and GFAP showed trends in biomarker ability to improve diagnosis, triage and clinical care in TBI in a wide range of contexts of care (emergency room, ward admission and ICU) and severities. GFAP, measured within 24 hours following mTBI, was found to improve the prediction of CT abnormalities.[18] Although the combination of GFAP and UCH-L1 did not enhance this performance, compared with GFAP alone,[18] it should be noted that GFAP and UCH-L1 were measured within 24 hours after TBI using a research-use only assay with poor agreement between replicates of biomarker assessments. No cut-off values for GFAP or UCH-L1 were obtained or predefined in the analysis. The present study will evaluate the diagnostic accuracy on the presence of CT findings of the combination of GFAP and UCH-L1, measured within 12 hours post TBI, using an automated VIDAS IVD-platform, in patients with mTBI. In addition, an external validation of the cut-off values for these two biomarkers will be performed. Finally, GFAP and UCH-L1 will be assessed regarding their possible prediction of mid-term neurological outcome and quality of life.

There are some limitations with the BRAINI protocol. First, variability in mTBI management and CT ordering may be expected in Spain, as there is no consensus regarding which clinical decision rules should be used for ordering CT, and inter-centre differences may exist in France as well. This may in turn influence the CT-positive prevalence across the sites. To understand the variability in CT ordering between and within countries and centres, the reason for performing cranial CT will be recorded and analysed in relation to the degree of compliance with clinical decision rules for CT ordering and the percentage of positive CTs in each centre. This information could help understand the generalisability of the results. Second, differences might occur between local and central CT readings. To mitigate this risk, an additional independent central CT reading will be performed for final adjudication. Third, only patients with mTBI with brain CT scans will be included in this study. Patients discharged from the ED following clinical examination without CT scan will not be included; therefore, their GFAP and UCH-L1 concentrations will not be captured. As a consequence, the value of GFAP and UCH-L1 in all mTBI presentations will not be determined.

## Author affiliations
[1]Department of Anaesthesia and Intensive Care, Univ. Grenoble Alpes, Centre Hospitalier Universitaire Grenoble Alpes, Grenoble Institut des Neurosciences, INSERM, U1216, Grenoble, France
[2]Servicio de Neurocirugía, Hospital Universitario 12 de Octubre, Universidad Complutense de Madrid, Instituto de Investigación imas12, Madrid, Spain
[3]bioMérieux, Clinical Unit, Chemin de l'Orme, Marcy l'Etoile, Spain
[4]Instituto de Investigación imas12, Hospital Universitario 12 de Octubre, SAMID, Madrid, France
[5]bioMérieux, Medical Affairs, Chemin de l'Orme, Marcy-l'Étoile, France

**Collaborators** Maxime Maignan, Emergency Department, University Hospital, Grenoble, France; Laurent Jacquin, Hospices Civils of Lyon, Emergency department, Edouard Herriot Hospital, Lyon, France; Marion Douplat, Hospices Civils de Lyon, Emergency Department, Lyon, France; Said Laribi, Department of Emergency Medicine, Tours University Hospital, Tours, France; Philippe Pes, Department of Emergency Medicine, University Hospital of Nantes, Nantes, France; Patrick Ray, Emergency Department, University Hospital of Dijon Bourgogne, Dijon, France; Jérémy Guenezan, CHU de Poitiers, Service des Urgences et SAMU 86, Poitiers, France, Université de Poitiers, Faculté de Médecine et de Pharmacie de Poitiers, Poitiers, France; Mustapha Sebbane, Department of Emergency Medicine and Prehospital Care, Montpellier University Hospital, Montpellier, France; Frédéric Balen, Emergency Department, Toulouse University Hospital, Toulouse, France; Guillaume Durand, Hôpital Nord-Ouest Villefranche sur Saône, France; Cordelia Abric, CH Annecy Genevois, FranceCédric Gil-Jardiné, Emergency Department, University Hospital Bordeaux, France; Podaru Mihai, Servicio de Urgencias, Hospital Universitario del Tajo, Madrid, Spain; Julian Morales, Servicio de Urgencias, Hospital Universitario Gregorio Marañón, Madrid, Spain; Ana Castuera, Servicio de Urgencias, Hospital Universitario Gregorio Marañón, Madrid, Spain; Mariana Garcia Ponce, Servicio de Neurocirugía, Instituto de Investigación imas 12, Hospital Universitario 12 de Octubre, Madrid, Spain; Maite Cuesta, Servicio de Neurocirugía, Instituto de Investigación imas 12, Hospital Universitario 12 de Octubre, Madrid, Spain; Jose A. F. Alén, Servicio de Neurocirugia, Hospital Universitario de la Princesa, Madrid, Spain.

**Contributors** All authors initiated the study design, wrote the study protocol and drafted the manuscript. All are part of the steering committee of the study. All authors attest to have significantly contributed to refinement of the study protocol and approved the final manuscript. All members of BRAINI contributed to the design and application of the protocol.

**Funding** This study is supported by a grant from the European Institute of Innovation and Technology (EIT) Health (BP 2019-2020). EIT Health is supported by EIT, a body of the European Union. bioMérieux is responsible for the development and manufacturing of the VIDAS GFAP and VIDAS UCHL-1 assays. bioMérieux will provide in-kind support to the study by supplying the assays for the measurement of UCH-L1 and GFAP necessary for this study.

**Competing interests** VB, OM and VP are employees of bioMérieux.

**Patient and public involvement** Patients and/or the public were not involved in the design, or conduct, or reporting, or dissemination plans of this research.

**Patient consent for publication** Not required.

**Provenance and peer review** Not commissioned; externally peer reviewed.

**ORCID iD**
Alfonso Lagares http://orcid.org/0000-0003-3996-0554

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
