## [Reviewer comments · BMJ Open]

ARTICLE DETAILS

TITLE (PROVISIONAL)	A study protocol for investigating the performance of an automated blood test measuring GFAP and UCH-L1 in a prospective observational cohort of patients with mild Traumatic BRAIN Injury: European BRAINI study
AUTHORS	Richard, Marion; Lagares, Alfonso; Bondanese, Victor; de la Cruz, Javier; Mejan, Odile; Pavlov, Vladislav; Payen, Jean-François

VERSION 1 – REVIEW

REVIEWER	Andrew I.R. Maas Emeritus Professor, dept of Neurosurgery Antwerp University Hospital
REVIEW RETURNED	20-Aug-2020

GENERAL COMMENTS	This manuscript reports a Protocol for an observational study in patients with mild TBI to evaluate the diagnostic and prognostic performance of an automated assay for the Biomarkers GFAP and UCH-L1. In total, 1760 patients will be recruited across 12 participating centers in France and Spain. Primary outcome is the presence of traumatic CT abnormalities on the initial CT Head. Secondary outcomes include neurological status and quality of life at 1 week and 3 months after injury. The Protocol is clearly described, and the evidence underpinning the study presented adequately. I have a few issues for clarification: 1.: IRB approval for the study has been obtained from local and national authorities in France and Spain, but I could not find any statement on informed consent. I would anticipate that informed consent would be required for this type of study given the use and storage of biological material (blood). Please clarify consent procedures – apologies if I may have missed it.2.: I note that 3 of the authors are employees of BioMerieux, the company that produces the automated assay. The funding statement states that the study is funded by a grant from the European Institute of Innovation and Technology (EIT) Health. No mention is made of any support or in-kind funding by BioMerieux. Please be explicit about any role of BioMerieux, also in case of in-kind support (for example providing the assays for free).3. The protocol states that previously published cut-off values for the biomarkers will be externally validated. Are you here referring to cut-off values published in the ALERT study, or do you have established cut-offs for the BioMerieux assay? If the former, how comparable are assay results between the 2 platforms?
---

REVIEWER	Brent Winston
-----------------	---------------

	Departments of Critical Care, Medicine and Biochemistry and Molecular Biology Cumming School Of Medicine University of Calgary Calgary, AB, Canada
REVIEW RETURNED	27-Aug-2020

GENERAL COMMENTS	Review: This study was designed to assesses the diagnostic accuracy of an automated assay (VIDAS-BTI assay) for the measurement of serum GFAP and UCH-L1 in a cohort of mTBI patients who received a CT scan as the standard of care. This is a prospective multicentre observational study of 1760 mTBI patients recruited in France and Spain across 12 participating sites. Adult patients with an initial Glasgow Coma Scale score of 13-15 and a brain CT-scan underwent blood sampling within 12 hours after TBI. The primary outcome measure is the diagnostic performance of an automated assay measuring serum concentrations of GFAP and UCH-L1 for discriminating between patients with positive and negative findings on brain CT-scans. Secondary outcome measures include the performance of these two biomarkers in predicting the neurological status and quality of life at 1 week and 3 months after the trauma. The authors describe the following strengths and limitations of the study  -This is a large prospective multicentre study to validate the value of serum biomarkers GFAP and UCH-L1 in predicting brain CT-scan findings after mTBI. -The additional value of these two biomarkers will be assessed regarding neurological outcome and quality of life. -Variability in the management of mTBI patients and in the indications of CT-scans may influence the rate of positive CT-scans in the different participating centres, which may have an impact on the statistical power of the study. Overall, this study could reveal important findings however there are some concerns and or suggestions highlighted below. Concerns and/or suggestions:  1) Although the authors recognize the problem of variability in managing indications for CT-scans, this is a critical point and may have a profound effect on the results. Is there any way a more standardized approach can be taken to request a CT-scan in order to remove the bias or variability that may occur for the request? This is clearly a regional effect with France being different than Spain. Clearly, criteria can be defined to include a CT-scan request. This could be a fundamental problem that could easily affect the results of the trial. 2) Is there a reference 'gold' standard measurement for GFAP and UCH-L1? Why not use the method used by the CENTRE-TBI study (E. Czeiter et al. / EBioMedicine 56 (2020) 102785) for comparison? In addition, there is no comparison with normal controls to standardize the measurement. Both of these may help define the actual value defined by the new measurement technique. Finally, there needs to be repeated measures using the new assay to see
---

	how repetitively precise the measurement is. 3) Is there going to be an analysis based on age and sex as the values of GFAP and UCH-L1 may vary with age and sex. 4) The authors state that “10-ml blood sample will be centrifuged without delay and frozen at -80°C.” There should be parameters placed on what ‘without delay’ means – ? within 2 h of sample collection? Samples left at room temperature for varied and extended periods of time may affect the actual measure of the biomarker of interest. Please define a time limit for processing and freezing the sample. 5) “In addition, an external validation of the cut-off values or these two biomarkers will be performed.” Please explain how this will be done. 6) There is value in doing a blood test in all patients with GCS 13-15 with TBI and then examining all those who get a CT-scan. This would enable one to determine how many have elevated brain injury biomarkers that were not CT-scanned. This would be a more complete analysis to examine how many biomarker positive patients were missed. 7) More discussion is needed on how this study is different from the CENTRE-TBI study (E. Czeiter et al. / EBioMedicine 56 (2020) 102785)? What will be the value added for this study?
--	--

VERSION 1 – AUTHOR RESPONSE

Reviewer: 1
 Reviewer Name
 Andrew I.R. Maas

Institution and Country
 Emeritus Professor,
 dept of Neurosurgery
 Antwerp University Hospital

Please state any competing interests or state ‘None declared’:
 None declared

Please leave your comments for the authors below
 This manuscript reports a Protocol for an observational study in patients with mild TBI to evaluate the diagnostic and prognostic performance of an automated assay for the Biomarkers GFAP and UCH-L1. In total, 1760 patients will be recruited across 12 participating centers in France and Spain. Primary outcome is the presence of traumatic CT abnormalities on the initial CT Head. Secondary outcomes include neurological status and quality of life at 1 week and 3 months after injury. The Protocol is clearly described, and the evidence underpinning the study presented adequately. I have a few issues for clarification:
 1.: IRB approval for the study has been obtained from local and national authorities in France and Spain, but I could not find any statement on informed consent. I would anticipate that informed consent would be required for this type of study given the use and storage of biological material (blood). Please clarify consent procedures – apologies if I may have missed it.

Response to Reviewer's comment 1.1:

In France the investigator has to detail the rights of the subject, collect the non-opposition of the patient to be enrolled in the study and report it in the patient medical file. If the patient doesn't want to participate in the study, his/her opposition also has to be documented in his/her medical file.

In Spain, the patient or next of kin (if the patient is not in condition of giving consent) has to sign a written informed consent before inclusion in the study.

Change in the manuscript 1.1: Study population subsection

Patients will be included after verification of the eligibility criteria. In France patient non-opposition to participate in the study must be documented. In Spain written informed consent will be obtained, before inclusion in the study, from the patient or next of kin if the patient is not in condition of giving consent.

2.: I note that 3 of the authors are employees of BioMérieux, the company that produces the automated assay. The funding statement states that the study is funded by a grant from the European Institute of Innovation and Technology (EIT) Health. No mention is made of any support or in-kind funding by BioMérieux. Please be explicit about any role of BioMérieux, also in case of in-kind support (for example providing the assays for free).

Response to Reviewer's comment 1.2:

We agree with this comment. We have clarified the co-funding contribution of bMx in the manuscript funding section.

Change in the manuscript 1.2: Funding

bioMérieux is responsible for the development and manufacturing of the VIDAS® GFAP and VIDAS® UCHL-1 assays. bioMérieux will provide in-kind support to the study by supplying the assays for the measurement of UCH-L1 and GFAP necessary for this study.

3. The protocol states that previously published cut-off values for the biomarkers will be externally validated. Are you here referring to cut-off values published in the ALERT study, or do you have established cut-offs for the BioMérieux assay? If the former, how comparable are assay results between the 2 platforms?

Response to Reviewer's comment 1.3:

Thank you for this comment. Accordingly, we have modified the statistical considerations section related to the external validation.

Change in the manuscript 1.3: Statistical considerations subsection

The Banyan BTI™ test for which Bayan biomarkers got FDA clearance is the only IVD reference assay for GFAP and UCHL-1 so far. It will be used as the reference for the VIDAS® GFAP and VIDAS® UCHL-1 assays.

We formulate the hypothesis that the automated assay measuring the serum concentrations of GFAP and UCH-L1 will have a sensitivity of at least 97% and an NPV of at least 99% to rule out the need for CT scan with accuracy. Assuming a prevalence of 11% of positive brain CT scans, the enrolment of

176 patients with positive CT-scans and 1424 patients with negative CT-scans is required to achieve the performances above mentioned. Assuming a 10% rate of loss to follow-up, the study recruitment target is set to 1760 patients across France and Spain.

Cut-off values for UCH-L1 (327 pg/ml) and GFAP (22 pg/ml) applied in the ALERT-TBI study were fixed for the Banyan BTI™ [12]. We will verify the predictive performances of UCH-L1 and GFAP measured on the VIDAS® instrument with these predefined cut-off values. An analysis on whether alternative cut-offs could provide better performance with the VIDAS® GFAP and VIDAS® UCH-L1, will also be performed based on BRAINI study data. A method comparison between Banyan BTI™ and VIDAS® VIDAS® GFAP combined with VIDAS® UCH-L1 will be performed and will contribute to optimal cut-offs selection.

Reviewer: 2

Reviewer Name

Brent Winston

Institution and Country

Departments of Critical Care, Medicine and Biochemistry and Molecular Biology

Cumming School Of Medicine

University of Calgary

Calgary, AB, Canada

Please state any competing interests or state 'None declared':

None declared

Please leave your comments for the authors below

Journal Review for BMJ Open

Manuscript ID: bmjopen-2020-043635

Article Type: Protocol

Title: Predictive performance of an automated assay for GFAP and UCH-L1 blood concentrations in mild Traumatic BRAIN Injury: European BRAINI study

Authors: Marion, R et al.

Review:

This study was designed to assesses the diagnostic accuracy of an automated assay (VIDAS-BTI assay) for the measurement of serum GFAP and UCH-L1 in a cohort of mTBI patients who received a CT scan as the standard of care.

This is a prospective multicentre observational study of 1760 mTBI patients recruited in France and Spain across 12 participating sites. Adult patients with an initial Glasgow Coma Scale score of 13-15 and a brain CT-scan underwent blood sampling within 12 hours after TBI. The primary outcome measure is the diagnostic performance of an automated assay measuring serum concentrations of

GFAP and UCH-L1 for discriminating between patients with positive and negative findings on brain CT-scans. Secondary outcome measures include the performance of these two biomarkers in predicting the neurological status and quality of life at 1 week and 3 months after the trauma.

The authors describe the following strengths and limitations of the study

-This is a large prospective multicentre study to validate the value of serum biomarkers GFAP and UCH-L1 in predicting brain CT-scan findings after mTBI.

-The additional value of these two biomarkers will be assessed regarding neurological outcome and quality of life.

-Variability in the management of mTBI patients and in the indications of CT-scans may influence the rate of positive CT-scans in the different participating centres, which may have an impact on the statistical power of the study.

Overall, this study could reveal important findings however there are some concerns and/or suggestions highlighted below.

Concerns and/or suggestions:

1) Although the authors recognize the problem of variability in managing indications for CT-scans, this is a critical point and may have a profound effect on the results. Is there any way a more standardized approach can be taken to request a CT-scan in order to remove the bias or variability that may occur for the request? This is clearly a regional effect with France being different than Spain. Clearly, criteria can be defined to include a CT-scan request. This could be a fundamental problem that could easily affect the results of the trial.

Response to reviewer comment 2.1.:

We agree. The study protocol has the advantage to reflect the use of the biomarker in clinical practice. Variability in the indication of CT in mild TBI is admitted in the literature. Although there is not a consensus regarding which guideline can be used in the emergency room for asking for a CT in Spain, this does not mean that no rule is followed. Criteria from two guidelines are used in combination (Canadian CT rule and New Orleans criteria). In French sites, decision on whether the CT is needed will be based on 2012 guidelines of the French Society of Emergency Medicine. However, it is planned in the protocol to record the reason to prescribe CT in order to measure the gap between guidelines and practice, and to assess its potential impact between and within French and Spanish centres.

Changes in the manuscript 2.1: Discussion section

There are some limitations with the BRAINI protocol. First, variability in mTBI management and CT ordering may be expected in Spain, as there is no consensus regarding which clinical decision rule should be used for ordering CT, and inter-centre differences may exist in France as well. This may in turn influence the CT-positive prevalence across the sites. To understand the variability in CT ordering between and within countries and centres, the reason for performing cranial CT will be recorded and analysed in relation to the degree of compliance with clinical decision rules for CT ordering and the percentage of positive CTs in each centre. This information could help understand the generalizability of the results.

2) Is there a reference 'gold' standard measurement for GFAP and UCH-L1? Why not use the method used by the CENTRE-TBI study (E. Czeiter et al. / EBioMedicine 56 (2020) 102785) for comparison? In addition, there is no comparison with normal controls to standardize the measurement. Both of these may help define the actual value defined by the new measurement technique. Finally, there needs to be repeated measures using the new assay to see how repetitively precise the

measurement is.

Response to reviewer comment 2.2.:

The method used by CENTRE-TBI study has been developed on the SimOa platform (Qanterix) for a research use only. The only reference assay for GFAP and UCH-L1 measurements is the Banyan BTI™ test for which Banyan biomarkers got FDA clearance in February 2018.

bioMérieux will perform analytical performance according to Clinical & Laboratory Standards Institute (CLSI) guidelines including precision study, i.e. repeatability and reproducibility of the assays. In the absence of reference standard and reference measurement procedure for GFAP and UCH-L1, the VIDAS® assays will be traceable to an internal standard. The metrological traceability chain defined in accordance with the standard ISO 17511 will ensure the GFAP and UCH-L1 values assigned to calibrators, product calibrator and patient samples.

Changes in the manuscript 2.2: Data analysis biomarkers:

bioMérieux will perform analytical performance according to Clinical & Laboratory Standards Institute (CLSI) guidelines including precision study, i.e. repeatability and reproducibility of the assays. In the absence of reference standard and reference measurement procedure for GFAP and UCH-L1, the VIDAS® assays will be traceable to an internal standard. The metrological traceability chain defined in accordance with the standard ISO 17511 will ensure the GFAP and UCH-L1 values assigned to calibrators, product calibrator and patient samples.

3) Is there going to be an analysis based on age and sex as the values of GFAP and UCH-L1 may vary with age and sex.

Response to the reviewers 2.3:

The distribution of biomarker levels will be analysed by age, sex and other predictors.

Changes in the manuscript 2.3: Statistical considerations.

The distribution of biomarker levels will be analysed by age, gender and other predictors.. As a secondary objective, the levels of GFAP and UCH-L1 obtained with the VIDAS® assays will be analysed according to the neurological outcome, the presence of posttraumatic symptoms and quality of life.

4) The authors state that “10-ml blood sample will be centrifuged without delay and frozen at -80°C.” There should be parameters placed on what ‘without delay’ means – ? within 2 h of sample collection? Samples left at room temperature for varied and extended periods of time may affect the actual measure of the biomarker of interest. Please define a time limit for processing and freezing the sample.

Response to reviewer comment 2.4:

We thank you for this comment. There is indeed a protocol for timely centrifuging the samples and storing them at -80°C. We have made changes in the manuscript accordingly.

Changes in the manuscript 2.4:

The 10-ml blood sample for determination of biomarkers will be collected using gel-separator tubes and allowed to clot for 30 minutes at room temperature (18-25°C) and then centrifuged at 2000G for 15 minutes. Serum will be aliquoted to 1ml cryovials, frozen and stored at -80°C within two hours of the extraction until shipment on dry ice to the central storage facility (bioMérieux R&D bioBank, Marcy

l'Étoile, France).

5) "In addition, an external validation of the cut-off values or these two biomarkers will be performed." Please explain how this will be done.

Response to reviewer comment 2.5.:

We have modified the statistical considerations section to address this comment.

Changes in manuscript 2.5:

Cut-off values for UCH-L1 (327 pg/ml) and GFAP (22 pg/ml) applied in the ALERT-TBI study were fixed for the Banyan BTI™ [12]. We will verify the predictive performances of UCH-L1 and GFAP measured on the VIDAS® instrument with these predefined cut-off values. An analysis on whether alternative cut-offs could provide better performance with the VIDAS® GFAP and UCH-L1, will also be performed based on BRAINI study data. A method comparison between Banyan BTI™ and VIDAS® GFAP combined with VIDAS® UCH-L1 will be performed and will contribute to optimal cut-offs selection.

6) There is value in doing a blood test in all patients with GCS 13-15 with TBI and then examining all those who get a CT-scan. This would enable one to determine how many have elevated brain injury biomarkers that were not CT-scanned. This would be a more complete analysis to examine how many biomarker positive patients were missed.

Response to reviewers comments 2.6.:

We thank you for this proposal. However, the objectives of our study are circumscribed to patients who need CT. Indeed, the number of patients with mild TBI and positive CTs are low (10%), even using pre-defined rules to perform CT. In other terms, these selected patients may have moderate-to-high probability of having a positive CT. The rate of missing CT positive patients following these rules is less than 5%. Because the objective of this study is to determine cut-off values for the VIDAS® GFAP and VIDAS® UCH-L1 to rule out positive CT scan, we have to select patients who fulfil clinical criteria for performing a CT.

We addressed this limitation in the last paragraph of the discussion: "Third, only mTBI patients with brain CT scans will be included in this study. Patients discharged from the ED following clinical examination without CT scan will not be included; therefore, their GFAP and UCH-L1 concentrations will not be captured. As a consequence, the value of GFAP and UCH-L1 in all TBI presentations will not be determined."

7) More discussion is needed on how this study is different from the CENTRE-TBI study (E. Czeiter et al. / EBioMedicine 56 (2020) 102785)? What will be the value added for this study?

Response to reviewers 2.7:

We acknowledge the importance of the study by E. Czeiter et al. However there are important differences with our study that will be added to the Discussion section:

(1) Czeiter's study is not focused on mild TBI patients arriving at ER since it recruited patients with any TBI severity at a wider range of healthcare contexts (ER only, ward admissions and ICU).

(2) The timing of blood sampling is up to 24 hours post-injury in Czeiter et al, versus within 12 hours in BRAINI.

(3) Czeiter's study assessed the concentration and relative performance of a panel of 6 biomarkers including UCH-L1 and GFAP. The concentration of UCH-L1 and GFAP was determined using the

Single Molecule Arrays technology (SiMoA) and the Neurology 4-plex A from Quanterix. Corp. (Lexington, MA). Quanterix technology is for research-use-only (RUO). This technology is not validated for clinical use.

(4) Although TBI severity is used to assess the discriminative ability of blood biomarkers for predicting the presence of CT abnormalities, no cut-off values have been determined or pre-defined for the biomarker assays in Czeiter's analysis.

(5) Although Czeiter's study shows trends in biomarker ability to improve diagnosis, triage and clinical care in TBI, BRAINI addresses more specifically the triage performance of two brain biomarkers using an automated platform to rule out the need for CT scan.

Changes in manuscript 2.7:

In the large CENTER-TBI cohort study, performance of a panel of six blood biomarkers including UCH-L1 and GFAP showed trends in biomarker ability to improve diagnosis, triage and clinical care in TBI in a wide range of contexts of care (Emergency Room, ward admission and ICU) and severities. GFAP, measured within 24 h following mTBI was found to improve the prediction of CT abnormalities [18]. Although the combination of GFAP and UCH-L1 did not enhance this performance, compared to GFAP alone [18], it should be noted that GFAP and UCH-L1 were measured within 24-h after TBI using a research-use only (RUO) assay with poor agreement between replicates of biomarker assessments. No cut-off values for GFAP or UCH-L1 were obtained or predefined in the analysis. The present study will evaluate the diagnostic accuracy on the presence of CT findings of the combination of GFAP and UCH-L1, measured within 12 hours post-TBI, using an automated VIDAS® IVD-platform, in mild TBI patients. In addition, an external validation of the cut-off values for these two biomarkers will be performed. Finally, GFAP and UCH-L1 will be assessed regarding their possible prediction of mid-term neurological outcome and quality of life.

VERSION 2 – REVIEW

REVIEWER	Andrew I.R. Maas Emeritus Professor, dept of neurosurgery, Antwerp University Hospital Belgium
REVIEW RETURNED	04-Nov-2020
GENERAL COMMENTS	In this revision, the reviewer comments have been addressed appropriately. I have no further comments.
REVIEWER	Brent Winston Departments of Critical Care, Medicine and Biochemistry and Molecular Biology Cumming School of Medicine University of Calgary Calgary, AB Canada
REVIEW RETURNED	26-Nov-2020
GENERAL COMMENTS	Review The authors have addressed the majority of this reviewer's concerns, however, there is still some concern about how the variability in managing indications for CT-scans between France and Spain, the inherent bias differences, may affect the results of the study if a single standard is not adopted in the protocol. The authors do explain that they will study the differences in CT-scan decisions. Although this is acceptable, the concern is it may affect the N

	needed to examine differences in decision making and ultimately defining who needs a CT scan.
--	---